# Underground laboratories

**Aldo Ianni**

I.N.F.N. Laboratori Nazionali del Gran Sasso, Via G. Acitelli 22 67100 Assergi (AQ), Italy

⋆ aldo.ianni@lngs.infn.it

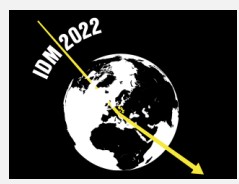 *14th International Conference on Identification of Dark Matter*
## Abstract

**A brief review on underground laboratories at the time of IDM 2022 is reported. General characteristics of these research infrastructures are discussed and a few highlights from different laboratories are reported. The idea of networking between underground laboratories is discussed.**

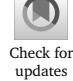
## 1 Introduction

Underground laboratories are research infrastructures built with a significant rock overburden to reduce the flux of muons induced by primary cosmic rays. The reduced ionizing radiation underground opens new possibilities to search for rare events, such as low energy (MeV scale) neutrino interactions, hypothetical dark matter particles interactions, and evidence of neutrinoless double beta decay. At present, there are 13 underground laboratories in operation worldwide. The map of these laboratories is shown in figure 1. The underground excavated volume and the muon flux are reported in figures 2 and 3, respectively. The total excavated volume is of the order of $1.5 \times 10^6$ m$^3$. The muon flux changes depending on the surface landscape: under a mountain the flux can have a strong dependence on zenith and azimuth angles. Figure 3 shows the difference of effective depth for a flat surface or a mountain profile.

Besides neutrino physics and astroparticle physics, other fields of research can benefit from the rock overburden that underground laboratories provide. Earth and environmental science, biology and plantery exploration, geophysics, and gravitational waves observation can be studied in these infrastructures. In the last decade these deep underground laboratories have been expanding research to neighboring sectors turning into multi-disciplinary research infrastructures although the main focus remains the rare events research.

Underground laboratories have many common features and at the same time they also have different specific characteristics (e.g. varying depths, geological compositions, laboratory sizes, access capabilities and support services provided) that make them more or less suitable for specific activities. For example, Boulby in the UK has surrounding geology that results in a particularly low ambient radon level underground; Canfranc in Spain and CallioLab in Finland have access to sites at different depths allowing studies with a different muons background level; JingPing in China and SNOlab in Canada are very deep with a significant reduction of the locally muon-induced background.

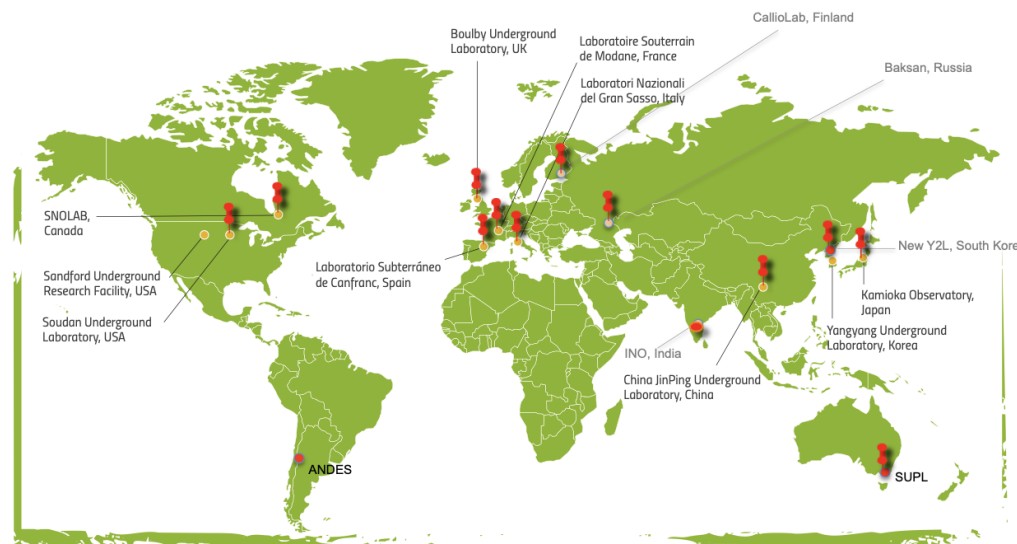

Figure 1: Deep underground laboratories around the world. Red thumbtacks correspond to underground laboratories discussed in this paper. Dots correspond to other laboratories and proposed projects.

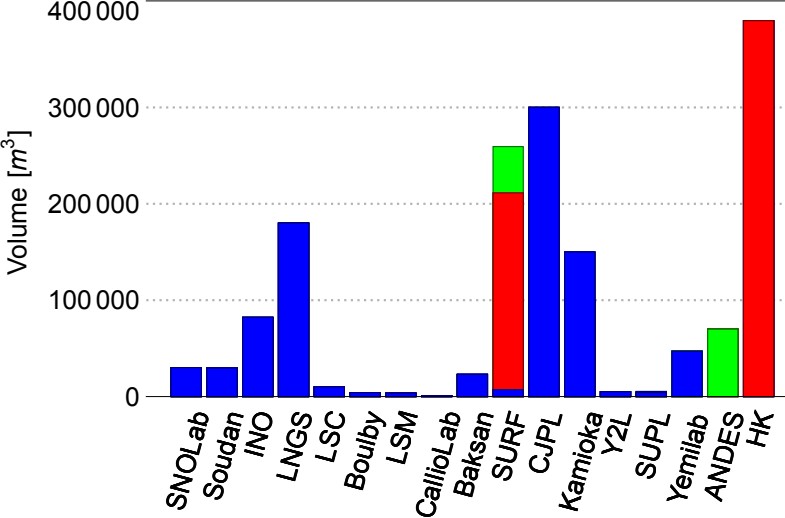

Figure 2: Excavated volume in underground laboratories. Blue: existing. Red: planned. Green: proposed.

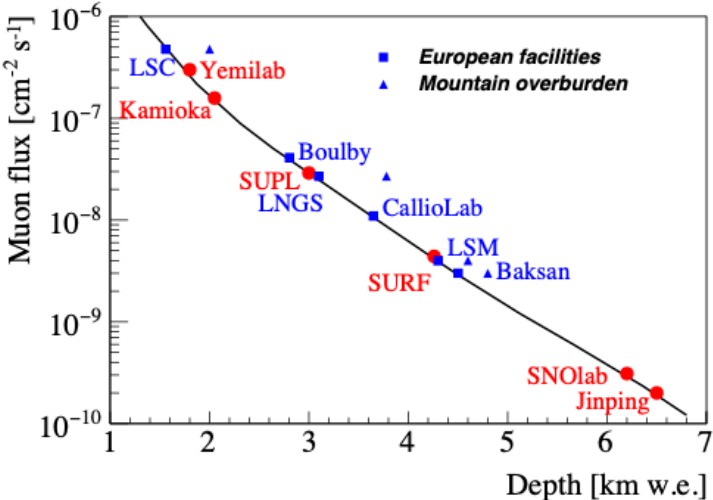

Figure 3: Muons flux in underground laboratories. The solid black line shows the muon flux as a function of depth for a flat overburden. For laboratories under a mountain the muon flux has an angular dependence. In the figure for these cases a maximum overburden is shown. This is greater than the overburden predicted under a flat surface. For comparison all laboratories have been overlaid to the solid line.

## 2 Facilities

Underground laboratories to support research can provide: (1) a unique environment for multi-disciplinary research; (2) effective radiation shielding; (3) above ground and underground support facilities, such as clean rooms and radon-free clean rooms, radio-purity assay equipment, cryogenic equipment; (4) material production and purification facilities. In the following we focus on some specific facilities provided by these infrastructures [1].

Radio-purity assay is a key technology for rare events research [2]. Underground laboratories have very high specialized equipment in this sector. We can count 76 high-purity germanium detectors (HPGe) for gamma spectroscopy which provide a unique material screening capability for the present and next-generation experiments. In addition, efforts are being made to push further the sensitivity of instruments to face new challenges for dark matter and neutrinoless double beta decay research. Radio-purity assay facilities also include high sensitivity radon detectors for gas and liquids [3, 4], beta and alpha spectrometers.

New technologies have been developed in underground laboratories. One example is the use of radon-free clean rooms with radon levels below 100 mBq/m$^3$ [4, 5]. These clean rooms are crucial for assembling dark matter detectors. Other examples of new technologies concern massive production of high radio-purity copper electro-formed, advanced technologies for cryogenic detectors, additive manufacturing, and innovative photo-detectors based on SiPM.

## 3 Highlights from underground laboratories

A few highlights from different underground laboratories are reported.

- LNGS [6]. For massive production of SiPM based photosensors a new facility, named NOA, is being built. NOA is a ISO6 Rn-free clean room with about 420 m$^2$ effective surface equipped with advanced instrumentation for packaging. In the framework of

DarkSide-20k NOA will provide about 20 m$^2$ instrumented surface with SiPM photo-detectors. The LUNA-MV infrastructure for nuclear astrophysics research is being commissioned. An advanced machining facility is being built. The Borexino detector is being decommissioned. COSINUS and DarkSide-20k are under construction.

- CJPL [7]. CJPL-II with order of 300k m$^3$ excavated volume is turning into a Deep Underground and ultra-low Radiation background Facility for frontier physics experiments (DURF) in 2024. DURF will be equipped with a crystal and copper electro-forming production facility, an assay radio-purity facility, and shielding infrastructures made of water and liquid nitrogen to host next-generation experiments.

- SURF [8]. LZ has provided first dark matter results [9]. The infrastructure for the long-baseline liquid argon project DUNE is being prepared. Outfitting of DUNE caverns is scheduled in 2024. In an early phase a so-called "module of opportunity" is planned in 50% of one DUNE cavern for temporary use of new projects. New short and long term planning for expansion of the underground laboratory are being studied (see figure 2).

- SNOLAB [10]. SuperCDMS first tower being installed by the end of 2022. SNO+ scintillator filling completed with 780 tons of LAB and 2.2 g/l of PPO. SNO+ will focus on background characterization and solar neutrino physics before turning to double beta decay physics with $^{130}$Te. Both nEXO and Legend-1T are under consideration for deployment at SNOlab.

- Kamioka [11]. KamLAND and Super-Kamiokande with Gd are taking data. The detector cavern for Hyper-Kamiokande is expected to be completed in 2024 and start of operation in 2027.

- LSC [12]. Improved results are reported from ANAIS [13]. NEXT-100 (double beta decay with $^{136}$Xe) construction phase has started. LSC becomes the reference laboratory for biology in underground in Europe. In addition an advanced facility for massive copper electro-formed production underground is being built [14].

- LSM [15]. A new under vault platform has been designed for the underground space. This enlarges the effective laboratory underground surface. LSM has the richest HPGe facility. DAMIC-M [16] becomes an important asset for LSM in the next future.

- Yemilab [17]. The new laboratory is starting operations in October 2022. Yemilab has a drive-in and vertical access. It will host COSINE-II and AMoRE-II.

- Boulby [18]. With its 4000 m$^3$ is the largest underground low radon environment with a cleanliness level in class ISO7 and ISO6 (in a sector). Boulby has a very diversified scientific program from astroparticle physics to planetary exploration. Plans are being developed for an important enlargement of the underground laboratory starting in 2030.

- SUPL [19]. The laboratory will start operations in December 2022. SUPL will be the first underground laboratory in the Southern Hemisphere after many years. The first experiment to be deployed in SUPL is SABRE-South [20].

## 4 Networking

It is understood that reinforcing connection between underground laboratories is a crucial asset. This effort is already underway in Europe, where five laboratories are already in operation.

Work load sharing and optimized used of facilities is a great advantage for next-generation experiments. A global network of underground laboratories will improve efficiency in responding to Collaborations requests. In Europe this connection includes a review of existing facilities and the establishment of a trans-national access policy. Working groups on specific topics are being organized between EU underground laboratories.

## 5 Conclusions

There are 13 underground laboratories with an outstanding and multi-disciplinary research program. The first laboratory in the Southern Hemispherebe will be opened by 2022. A second one, ANDES, is under consideration. Key technologies have been developed in these infrastructures to support research. The need for a robust coordination and global connection between underground laboratories is becoming a crucial asset for next-generation experiments.

## Acknowledgements

Acknowledgements to Sean Pauling (Boulby), Carlos Pena-Garay (LSC), Jules Gascon(LSM), Jari Joutsenvaara/Julia Puputti (CallioLab), Ezio Previtali (LNGS), Kim Yeongduk/Douglas Leonard(Y2L and Yemilab), Elisabetta Barberio(SUPL), Richard Ford(SNOLAB), Jaret Heise(SURF), Zhi Zeng(CJPL), Shigetaka Moriyama (Kamioka), Xavier Bertou (ANDES).

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
