# Peer review of "Underground laboratories"

_SciPost Physics Proceedings, doi:SciPost Phys. Proc. 12, 007 (2023)_

## Round 1 · Referee Report · Anonymous (Referee 1) · 2022-10-19

Report
In this proceeding author does a brief review of underground laboratories at the time of the conferences and the status of them. The manuscript is clearly written and well organised. It is also suitably formatted for publication.
I recommend the manuscript for publication with some recommendations.
I recommend the manuscript for publication with some recommendations.
Requested changes
- Fig 1 should be explained. The difference between black and grey is not strong enough.
- Confusion through the paper - how many labs are there.
- P.2 “..some 76….” Some or 76?
- Be consistent “electro-forming” or “e-forming”
- P.3 “NOW is a ISO6 Rn-free clean…” -> NOA?
- Add links to the labs (websites, papers or so)
- Fig.2 15 existing labs, in introduction 13.
- More explanations for Fig.3
- Ref 7-10 provide other references if possible.

---

## Editorial Decision

published